# European Registry on *Helicobacter pylori* Management: Effectiveness of First and Second-Line Treatment in Spain

**DOI:** 10.3390/antibiotics10010013

**Published:** 2020-12-25

**Authors:** María Caldas, Ángeles Pérez-Aisa, Manuel Castro-Fernández, Luis Bujanda, Alfredo J. Lucendo, Luis Rodrigo, Jose M. Huguet, Jorge Pérez-Lasala, Javier Molina-Infante, Jesús Barrio, Luis Fernández-Salazar, Ángel Lanas, Mónica Perona, Manuel Domínguez-Cajal, Juan Ortuño, Blas José Gómez-Rodríguez, Pedro Almela, Josep María Botargués, Óscar Núñez, Inés Modolell, Judith Gómez, Rafael Ruiz-Zorrilla, Cristóbal De la Coba, Alain Huerta, Eduardo Iyo, Liliana Pozzati, Rosario Antón, Mercé Barenys, Teresa Angueira, Miguel Fernández-Bermejo, Ana Campillo, Javier Alcedo, Ramón Pajares-Villaroya, Marianela Mego, Fernando Bermejo, José Luis Dominguez-Jiménez, Llúcia Titó, Nuria Fernández, Manuel Pabón-Carrasco, Ángel Cosme, Pilar Mata-Romero, Noelia Alcaide, Inés Ariño, Tommaso Di Maira, Ana Garre, Ignasi Puig, Olga P. Nyssen, Francis Megraud, Colm O’Morain, Javier P. Gisbert

**Affiliations:** 1Gastroenterology Unit, Hospital Universitario de La Princesa, Instituto de Investigación Sanitaria Princesa (IIS-IP), Universidad Autónoma de Madrid (UAM) and Centro de Investigación Biomédica en Red de Enfermedades Hepáticas y Digestivas (CIBERehd), 28006 Madrid, Spain; maria.caldas@salud.madrid.org (M.C.); anagarre.laprincesa@gmail.com (A.G.); opn.aegredcap@aegastro.es (O.P.N.); 2Digestive Unit, Agencia Sanitaria Costa del Sol, Red de Investigación en Servicios de Salud en Enfermedades Crónicas (REDISSEC), 29651 Marbella, Spain; drapereza@hotmail.com (Á.P.-A.); nuriafm@hcs.es (N.F.); 3Department of Gastroenterology, Hospital de Valme, 41014 Sevilla, Spain; manuel.castro.sspa@juntadeandalucia.es; 4Department of Gastroenterology, Hospital Donostia/Instituto Biodonostia and CIBERehd, Universidad del País Vasco (UPV/EHU), 20014 San Sebastián, Spain; luis.bujanda@osakidetza.net (L.B.); acosme@chdo.osakidetza.net (Á.C.); 5Department of Gastroenterology, Hospital General de Tomelloso and CIBERehd, 13700 Ciudad Real, Spain; ajlucendo@sescam.jccm.es (A.J.L.); teresa.anmtangueira@sescam.jccm.es (T.A.); 6Gastroenterology Unit, Hospital Central de Asturias, 33011 Oviedo, Spain; lrrodrigo@uniovi.es; 7Gastroenterology Unit, Consorcio Hospital General Universitario de Valencia, 46014 Valencia, Spain; huguet_jos@gva.es; 8Digestive Service, HM Sanchinarro, 28050 Madrid, Spain; jperez@hmhospitales.com; 9Department of Gastroenterology, Hospital San Pedro de Alcántara and CIBERehd, 10003 Cáceres, Spain; javier.molinai@salud-juntaex.es (J.M.-I.); pilar.matar@salud-juntaex.es (P.M.-R.); 10Department of Gastroenterology, Hospital Universitario Río Hortega, 47012 Valladolid, Spain; jbarrio@saludcastillayleon.es; 11Digestive Service, Hospital Clínico Universitario de Valladolid, 47003 Valladolid, Spain; lfernandezsa@saludcastillayleon.es (L.F.-S.); nalcaide@saludcastillayleon.es (N.A.); 12Digestive Service, Hospital Clínico Universitario Lozano Blesa and CIBERehd, 50009 Zaragoza, Spain; alanas@unizar.es (Á.L.); iarinnop@salud.aragon.es (I.A.); 13Gastroenterology Unit, Hospital Quirón Marbella, 29603 Málaga, Spain; monica.perona@quiron.es; 14Digestive Service, Hospital General San Jorge, 22004 Huesca, Spain; mdominguezc@salud.aragon.es; 15Digestive Service, Hospital Universitari y Politècnic La Fe de Valencia and CIBERehd, 46026 Valencia, Spain; jortunoc@comv.es (J.O.); dimaira_tom@gva.es (T.D.M.); 16Digestive Service, Hospital Universitario Virgen Macarena, 41009 Sevilla, Spain; gomezblasj@gmail.com; 17Digestive Service, Hospital Universitari General de Castelló, 12004 Castellón, Spain; almela_ped@gva.es; 18Digestive Service, Hospital Universitari de Bellvitge, 08907 Barcelona, Spain; jbotargues@bellvitgehospital.cat; 19Digestive Service, Hospital Universitario Sanitas La Moraleja, 28050 Madrid, Spain; onumar@gmail.com; 20Digestive Service, Consorci Sanitari de Terrassa, 08191 Barcelona, Spain; ines.modolell@gmail.com; 21Digestive Service, Complejo Asistencial Universitario de Burgos, 09006 Burgos, Spain; jgomezcam@saludcastillayleon.es; 22Digestive Service, Hospital Sierrallana, 39300 Cantabria, Spain; rafael.ruiz@scsalud.es; 23Digestive Service, Hospital de Cabueñes, 33394 Asturias, Spain; delacoba76@hotmail.com; 24Digestive Service, Hospital de Galdakao-Usansolo, 48960 Vizcaya, Spain; ALAIN.HUERTAMADRIGAL@OSAKIDETZA.EUS; 25Digestive Service, Hospital Comarcal de Inca, 07300 Mallorca, Spain; eduardoy.iyo@hcin.es; 26Digestive Service, Hospital de Mérida, 06800 Badajoz, Spain; santelmo0054@hotmail.com; 27Digestive Medicine Department, Hospital Clínic Universitari de Vàlencia, 46010 Valencia, Spain; M.Rosario.Anton@uv.es; 28Digestive Service, Hospital de Viladecans, 08840 Barcelona, Spain; mbarenys@gencat.cat; 29Digestive Service, Clínica San Francisco, 10002 Cáceres, Spain; miguel.fernandez@salud-juntaex.es; 30Digestive Service, Hospital Reina Sofía, Tudela, 31500 Navarra, Spain; ana.campillo.arregui@navarra.es; 31Digestive Service, Hospital de Barbastro, 22300 Huesca, Spain; jalcedo@salud.aragon.es; 32Digestive Service, Hospital Infanta Sofía, 28703 Madrid, Spain; ramon.pajares@salud.madrid.org; 33Digestive Service, Hospital Universitario General de Catalunya, 08195 Barcelona, Spain; marianela.mego@quironsalud.es; 34Digestive Service, Hospital Universitario de Fuenlabrada, idiPAZ, 28942 Madrid, Spain; fernando.bermejo@salud.madrid.org; 35Digestive Service, Hospital Alto del Guadalquivir, 23740 Jaén, Spain; jldominguez@ephag.es; 36Digestive Service, Hospital de Mataró, 08304 Barcelona, Spain; ltito@csdm.cat; 37Digestive Service, Centro Universitario de Cruz Roja, Universidad de Sevilla, 41004 Sevilla, Spain; mpabon@cruzroja.es; 38Digestive Service, Althaia Xarxa Assistencial Universitària de Manresa and Universitat de Vic-Universitat Central de Catalunya (UVicUCC), 08242 Manresa, Spain; ipuig@althaia.cat; 39Laboratoire de Bactériologie, Hôpital Pellegrin, Bordeaux & INSERM U1053 BaRITOn, Université de Bordeaux, 33076 Bordeaux, France; francis.megraud@chu-bordeaux.fr; 40Department of Clinical Medicine, Trinity College Dublin, D24 NR0A Dublin, Ireland; ColmOMorain@rcpi.ie

**Keywords:** *Helicobacter pylori*, treatment, first-line, second-line, Spain

## Abstract

The management of *Helicobacter pylori* infection has to rely on previous local effectiveness due to the geographical variability of antibiotic resistance. The aim of this study was to evaluate the effectiveness of first and second-line *H. pylori* treatment in Spain, where the empirical prescription is recommended. A multicentre prospective non-interventional registry of the clinical practice of European gastroenterologists concerning *H. pylori* infection (Hp-EuReg) was developed, including patients from 2013 until June 2019. Effectiveness was evaluated descriptively and through a multivariate analysis concerning age, gender, presence of ulcer, proton-pump inhibitor (PPI) dose, therapy duration and compliance. Overall, 53 Spanish hospitals were included, and 10,267 patients received a first-line therapy. The best results were obtained with the 10-day bismuth single-capsule therapy (95% cure rate by intention-to-treat) and with both the 14-day bismuth-clarithromycin quadruple (PPI-bismuth-clarithromycin-amoxicillin, 91%) and the 14-day non-bismuth quadruple concomitant (PPI-clarithromycin-amoxicillin-metronidazole, 92%) therapies. Second-line therapies were prescribed to 2448 patients, with most-effective therapies being the triple quinolone (PPI-amoxicillin-levofloxacin/moxifloxacin) and the bismuth-levofloxacin quadruple schemes (PPI-bismuth-levofloxacin-amoxicillin) prescribed for 14 days (92%, 89% and 90% effectiveness, respectively), and the bismuth single-capsule (10 days, 88.5%). Compliance, longer duration and higher acid inhibition were associated with higher effectiveness. “Optimized” *H. pylori* therapies achieve over 90% success in Spain.

## 1. Introduction

*Helicobacter pylori* (*H. pylori*) is a gram-negative bacterium with an estimated prevalence in southern Europe of 50%, involved in several important diseases such as chronic gastritis, peptic ulcer disease and gastric cancer, as well as in some important extra-gastric diseases such as iron deficiency anaemia or idiopathic thrombocytopenic purpura, among others [1,2,3,4,5]. These two factors are enough to explain the importance of finding a successful therapy able to eliminate the bacterium with the least amount of antibiotic treatment attempts. This is especially important considering the effect of antibiotics on the patient’s gut microbiota and the emergence of multi-resistant bacterial strains worldwide [6,7,8].

However, an effective global treatment has not been found. Although several factors might be considered, the geographical variability of antibiotic resistance among different *H. pylori* strains (due to previous antibiotic exposure of both the patient and the community) represents an important obstacle, especially considering the extended recommendation of using empirical prescriptions [1,9,10]. In this sense, clarithromycin, metronidazole and levofloxacin, antibiotics frequently used against *H. pylori*, have shown at least moderate resistance rates in southern Europe and specifically in Spain, which has justified the emergence of numerous and varied optimization strategies [9,11,12].

With the aim of obtaining updated information on *H. pylori* treatment in Spain and to find strategies to improve therapies in this area, we designed this long-term prospective clinical practice study.

## 2. Results

### 2.1. Baseline Characteristics

In total, 53 Spanish hospitals were selected (Appendix A) and 14,128 patients were included for the descriptive demographic analysis (Table 1).

### 2.2. Treatment Use and Effectiveness

#### 2.2.1. First-Line Treatment

A total of 10,267 patients received an empirical first-line treatment. The most frequently prescribed regimens were: non-bismuth quadruple concomitant therapy (PPI-clarithromycin-amoxicillin-metronidazole, all four drugs administered concomitantly, *n* = 4051, 40%), standard triple regimen (PPI-clarithromycin-amoxicillin, *n* = 2712, 26%), bismuth quadruple therapy (bismuth single-capsule, marketed as Pylera^®^ (Allergan, Inc Dublin, IE) containing tetracycline-metronidazole-bismuth salts administered together with a PPI, *n* = 1660, 16%), bismuth-clarithromycin quadruple therapy (PPI-clarithromycin-amoxicillin-bismuth, *n* = 1055, 10%), non-bismuth sequential quadruple regimen (PPI-amoxicillin during five days, followed by PPI-clarithromycin-metronidazole during the five remaining days, *n* = 230, 2.2%) and clarithromycin-metronidazole triple therapy (PPI-clarithromycin-metronidazole, *n* = 124, 1.2%). The prescription trends over time, as shown in Figure 1.

Overall effectiveness of the first-line therapies reached 88% on the mITT analysis. The highest effectiveness was obtained with the bismuth single-capsule (95%), the bismuth-clarithromycin therapy (91%) and the concomitant therapy (90%) (Table 2). Adverse events (AE) appeared in 25% of the patients; however, most of them were of mild intensity (62%) and only 0.2% of the patients presented a serious AE. Compliance was higher than 97% (see Appendix A for a detailed analysis of safety).

The overall multivariate analysis performed with the first-line treatment showed that higher effectiveness was associated with good compliance (OR = 4.1; 95% CI: 3.0–5.5), extended therapies (10 days (OR = 4.5; 95% CI: 3.2–6.2) or 14 days length (OR = 4.1; 95% CI: 2.9–5.9)), higher gastric acid inhibition (standard (OR = 1.4; 95% CI: 1.2–1.7) or high PPI doses (OR = 2.1; 95% CI: 1.7–2.4)), presence of peptic ulcer (OR = 1.2; 95% CI: 1.0–1.5) and male gender (OR = 1.2; 95% CI: 1.1–1.4) (see Table 3 for the analysis of each therapy).

#### 2.2.2. Second-Line Treatment

A total of 2448 patients received an empirical second-line therapy. Five therapies were most frequently used: the levofloxacin-amoxicillin triple therapy (PPI-amoxicillin-levofloxacin, *n* = 944, 39%), the bismuth-levofloxacin quadruple therapy (PPI-amoxicillin-levofloxacin-bismuth, *n* = 475, 19.4%), the bismuth single-capsule (*n* = 454, 18.6%), the moxifloxacin-amoxicillin therapy (PPI-amoxicillin-moxifloxacin, *n* = 136, 5.6%) and the concomitant therapy (*n* = 121, 4.9%). Treatment prescriptions over time are depicted in Figure 1.

Overall effectiveness of second-line therapies reached 84% on the mITT analysis. Highest effectiveness was obtained with the bismuth-levofloxacin therapy (89%) and the bismuth single-capsule (88%) (Table 2). AE were reported in 28% of the cases (49% and 47% of them being of moderate and mild intensity, respectively, and only one patient showing a serious AE). Compliance was higher than 95% (see Appendix A for a detailed analysis of safety).

The overall multivariate analysis performed with second-line treatment showed that higher effectiveness was associated with good compliance (OR = 3.4; 95% CI: 1.7–6.9), high PPI dose (OR = 1.9; 95% CI: 1.4–2.6) and 14-day therapy (OR = 1.5; 95% CI: 1.1–2.1) (See Table 4 for the analysis of separated therapies considering only the three therapies more frequently used in second-line in Spain).

Effectiveness by treatment duration and PPI doses in first- and second- line regimens are shown in Appendix A.

### 2.3. Penicillin Allergic Patients

A total of 411 patients allergic to penicillin received an empirical first-line therapy, being the most frequent, the bismuth single-capsule (*n* = 154, 37.5%) and the clarithromycin-metronidazole therapy (*n* = 117, 28.5%). The overall effectiveness was 81% in first-line, although the bismuth single-capsule reached 94% mITT effectiveness. A second-line attempt was empirically used in 137 patients allergic to penicillin, being the bismuth single-capsule (*n* = 34, 24.8%), the most frequently used.

Results on effectiveness, safety and compliance as well as effectiveness stratified by length and PPI dose, are shown in Appendix A.

## 3. Discussion

Treatment of *H. pylori* infection in Spain in first- and second- line, in which the empirical approach is generally recommended, still remains a challenge, especially considering the more demanding threshold of effectiveness required lately (90%) [6]. This has led to the progressive complexity of the regimens prescribed, which involve an increasing use of quadruple regimens, longer prescriptions (10–14 days) and higher PPI doses over time [1,9].

Concerning first-line treatment, we found an overall effectiveness of 88% in our cohort, close to the optimal threshold required, combined with an optimal safety profile [6,13].

Analysis of each specific therapy revealed that the standard triple therapy containing clarithromycin and amoxicillin, recommended throughout several years in Spain, only reached approximately 80% success, in line with previous evidence [14]. Its low effectiveness, together with the increasing clarithromycin resistance rates over time (up to 20% recently notified) led to the discontinuation of its use in our area [6,9,12]. This drop in prescriptions over time in our cohort brought with it the increase of use of quadruple therapies, such as the concomitant, the bismuth single-capsule and the bismuth-clarithromycin therapies, which also were, precisely, the therapies showing the highest effectiveness (≥90%).

Concomitant treatment was the therapy most frequently used in first-line, in agreement with recommendations provided by national guidelines [9]. This therapy showed 90% effectiveness, similar to previous reports, and was mostly due to the relatively low dual resistance rates to both clarithromycin and metronidazole documented in our country [9,12,15,16,17]. With regard to the bismuth-clarithromycin quadruple therapy, a progressive rise on its prescription was seen over time, and it showed a very high effectiveness (91%), similar to previous evidence [18]. In fact, data coming from areas with higher clarithromycin resistance than Spain such as China still show high success of this therapy, which is thought to be at least partially compensated by the use of bismuth [19]. This is encouraging, assuming a hypothetical increase of clarithromycin resistance in Spain in the following years. The use of a bismuth quadruple therapy that contains tetracycline and metronidazole also avoids the problem of clarithromycin resistance due to the absence of this antibiotic and the low probability of developing resistance to components such as bismuth or tetracycline by *H. pylori* [9,20]. Moreover, the commercialization of a pill containing the three aforementioned drugs facilitated the access to tetracycline, which used to be hardly available in this area [9,21]. This therapy showed an eradication rate of 95% in our cohort, similar to what was described in a recently published meta-analysis [22].

When we analysed the variables associated with the increase of effectiveness, good compliance (≥90% of drug intake) showed the highest association for overall treatment and also for nearly all the therapies analysed individually. Other authors had previously reported this association [17,18,23,24]. Therefore, strategies designed to pursue the best compliance possible are needed.

The use of standard PPI doses (≈40 mg omeprazole equivalents b.i.d.) or high PPI doses (≈60 mg omeprazole equivalents b.i.d.) showed higher effectiveness in our cohort in the overall analysis and in the concomitant, bismuth-clarithromycin and bismuth single-capsule therapies. Previous reports had already shown a beneficial effect of higher grades of gastric acid inhibition on the eradication treatment, especially in amoxicillin-containing therapies, considering the higher susceptibility of *H. pylori* to amoxicillin in a less acidic environment. This is particularly important for the treatment of clarithromycin-resistant strains [17,18,24,25]. The finding of this increase in effectiveness with the bismuth single-capsule is remarkable, as other authors reported an absence of association [22]. Although an adjuvant effect of high PPI doses in metronidazole resistant strains has been suggested, specific studies on this issue are strongly encouraged [26].

Increasing the length of therapy (10 or 14 days instead of 7) also increased overall effectiveness in our first-line cohort. The three quadruple therapies showing highest effectiveness were compared between 10 and 14-days (7-day treatment was excluded from analysis in these therapies because of the small number of patients included). While the concomitant therapy showed a tendency towards better results with the longest duration in accordance with previous reports [15], neither of the two remaining quadruple therapies (both containing bismuth) showed association between treatment length and effectiveness in our study. However, it is important to clarify that prescriptions were made for 10 days in the bismuth single-capsule and for 14 days in the bismuth-clarithromycin therapy in 99% of the cases, which could have limited the comparisons.

The overall effectiveness of second-line attempts was 84% in our cohort, close but not reaching the 90% of threshold previously mentioned. The prescription of second-line therapies in our cohort has evolved over time similarly to what happened in the first-line: in this case, quinolone-containing triple therapies experienced a decrease of use while quadruple regimens experienced the opposite (i.e., bismuth-levofloxacin quadruple therapy or the bismuth single-capsule) [9].

The levofloxacin-amoxicillin triple therapy has been one of the most traditionally employed regimens in rescue attempts in Spain, based on the previously considered “acceptable” results (showing around 80% effectiveness in our cohort) [27]. Although this rate is currently considered unacceptable, the increase of treatment duration up to 14 days led to a 92% effectiveness (similar to the findings obtained in areas with levofloxacin resistance rates similar to Spain) making it still an interesting alternative to consider [28]. Something similar happened with the moxifloxacin-amoxicillin therapy, which showed 91% effectiveness in our cohort, a figure markedly higher than that reported by other authors (78–82%) [29,30]. These studies included high doses of PPI and a 14-day duration, similar to the majority of the prescriptions made in our cohort. Updated evidence concerning both quinolone-containing triple therapies is needed before definitive conclusions are drawn.

However, triple levofloxacin-based therapies suffer the limitation of the increasing resistance to quinolones worldwide [11]. In order to overcome this resistance, bismuth has been added to the levofloxacin-amoxicillin therapy, forming the quadruple therapy previously described. This therapy achieved around 90% effectiveness in our cohort, in accordance with previous studies [31,32]. It is important to remark that more than 90% of the prescriptions had 14-day lengths and used high PPI doses, which could have increased the effectiveness per se. In addition, it is important to remember that the local cut-off point of quinolone resistance for cost-effectiveness of this quadruple therapy has yet to be established.

The use of bismuth single-capsule has provided effectiveness around 90% concerning rescue attempts, representing an interesting alternative rescue-treatment to consider [22]. More than 98% of the prescriptions in our study were made for a 10-day duration, following the currently recommended dosage.

In our cohort, the effectiveness of clarithromycin-containing triple therapies with either metronidazole or levofloxacin was suboptimal in the allergic to penicillin population, in agreement with previously published evidence [33]. The use of the bismuth single-capsule in this setting has markedly increased effectiveness, achieving up to 94% in our cohort in first-line treatment, thus confirming this regimen as the current therapy of choice in this population [1,9,34].

Despite these relevant findings, our study has several limitations. The main one is the low proportion of patients subjected to culture testing. It is unquestionable that promoting the creation of big-size databases to update local antibiotic resistance information could improve effectiveness. Nonetheless, the current study was developed to evaluate the routine practice of gastroenterologists in Spain, where the empirical prescription is generally advocated up to three eradication attempts [9]. Another limitation of the present study is its observational design, with the consequent potential higher risk of bias. However, the large size of our sample (including more than 10,000 patients in first line) should have compensated for this limitation. Due to the source of the data (concerning real updated information from routine clinical practice in Spain), our study provides a comprehensive global overview of the current local management, whose conclusions are easily applicable in routine practice.

## 4. Materials and Methods

This study is a sub-analysis focused on the Spanish centres actively participating in the “European Registry on *H. pylori* Management” (Hp-EuReg), an international (27 countries), multicentre (300 investigators), prospective non-interventional registry that started in 2013 and promoted by the European Helicobacter and Microbiota Study Group (www.helicobacter.org) [35]. The established Scientific Committee, national coordinators, gastroenterologist recruiting investigators and a list of variables and outcomes are detailed in the previously published protocol [36]. This protocol was approved by the Research Ethics Committee of La Princesa University Hospital (Madrid, Spain) and prospectively registered at ClinicalTrials.gov (NCT02328131). Written informed consent was obtained from each patient included in the study.

Data were recorded in an Electronic Case Report Form (e-CRF), collected and managed using REDCap, and were subjected to quality review. REDCap is an electronic data capture tool hosted at “Asociación Española de Gastroenterología” (AEG; www.aegastro.es), a non-profit scientific and medical society focused on gastroenterology research [37,38].

The aim of the current sub-analysis was to evaluate in the Hp-EuReg the effectiveness of first and second-line treatments in Spain. Secondary aims included the evaluation of strategies designed to increase effectiveness, prescription trends over time, and safety.

### 4.1. Variables

The e-CRF registered 290 variables including demographics, comorbidity, data on infection and diagnosis, previous eradication attempts, current treatment, compliance, AE and effectiveness [36]. The variable treatment length was assessed using three categories for first-line therapies, corresponding with the most frequent treatment durations: 7, 10 and 14 days; and two categories for second-line attempts: 10 and 14 days. The variable dose of the proton pump inhibitor (PPI) was grouped in three categories (low, standard and high doses) according to reports by Graham and Kirchheiner [25,39] (See Appendix A).

### 4.2. Effectiveness Analysis

*H. pylori* eradication was confirmed with at least one of the following diagnostic methods: urea breath test, stool antigen test and/or histology; at least one month after completing eradication treatment. The treatment eradication rate was the main outcome and was studied in three sets of patients as follows: intention-to-treat (ITT) analysis included all patients registered up to June 2019, to allow at least a 6-month follow-up, and lost to follow-up cases were considered treatment failures. Per-protocol (PP) analysis included all cases that finished follow-up and had taken at least 90% of the treatment drugs. A modified ITT (mITT) analysis was designed aiming to reach the closest result to clinical practice, including all cases that had completed follow-up (that is, a confirmatory test of success or failure was available after eradication treatment).

### 4.3. Statistical Analysis

Continuous variables are presented as the arithmetic mean and standard deviation. Qualitative variables are shown as percentages. Differences between groups were analysed with the Chi-square test. Multivariate analysis was performed using a logistic regression model by means of the stepwise forward likelihood method with *H. pylori* mITT eradication as dependent variable and age, gender, treatment duration, PPI dose, compliance, and previous use of clarithromycin as independent factors (this last one only in second-line approaches). Significance was considered at *p* < 0.05.

### 4.4. Outcome Reporting

Analyses were performed separately in the global population, first- and second-line therapy samples and penicillin-allergic patients. Effectiveness, safety and compliance were evaluated in those patients receiving empirical therapies (not culture-guided) (Figure 2).

## 5. Conclusions

In Spain, the standard triple therapy containing clarithromycin and amoxicillin shows an inadequate effectiveness; therefore, its empirical use should be abandoned. The best effectiveness in first-line was obtained with the bismuth single-capsule prescribed for 10 days, and the concomitant and bismuth-clarithromycin quadruple therapies, both prescribed for 14 days. In second-line, the best effectiveness was obtained with a 10-day bismuth single-capsule regimen, and also with 14-day quinolone-containing therapies, either with or without bismuth. In general terms, we have seen a tendency over time towards the rise in the use of quadruple therapies, longer duration regimens and higher doses of PPIs. This is in line with current recommendations by the main guidelines, which are, slowly but steadily, being implemented by Spanish gastroenterologists in clinical practice. These three factors, together with good compliance, seem to be, in general terms, the four main strategies to increase effectiveness.

## Figures and Tables

**Figure 1 antibiotics-10-00013-f001:**
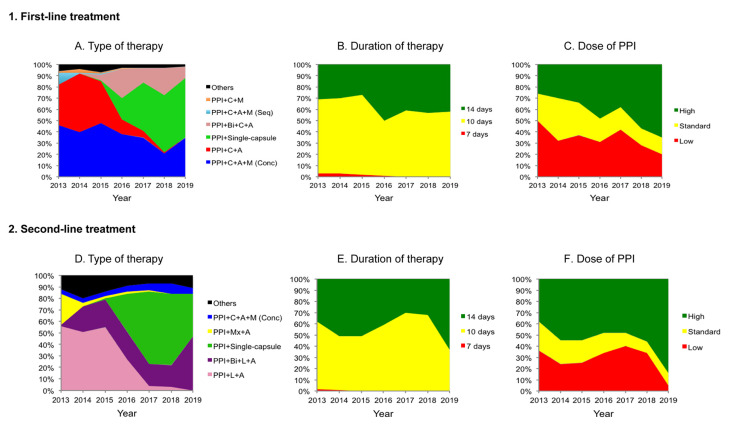
Prescription trends in first- (**A**) Type of therapy; (**B**) Duration of therapy; (**C**) Dose of PPI and second-line (**D**) Type of therapy; (**E**) Duration of therapy; (**F**) Dose of PPI treatment. PPI: proton-pump inhibitor, C: clarithromycin, A: amoxicillin, M: metronidazole, Single-capsule: three-in-one single capsule, Bi: bismuth, Conc: concomitant administration, Seq: sequential administration, L: levofloxacin, Mx: Moxifloxacin, Low: ≈20 mg omeprazole equivalents b.i.d., Standard: ≈40 mg omeprazole equivalents b.i.d., High: ≈60 mg omeprazole equivalents b.i.d.

**Figure 2 antibiotics-10-00013-f002:**
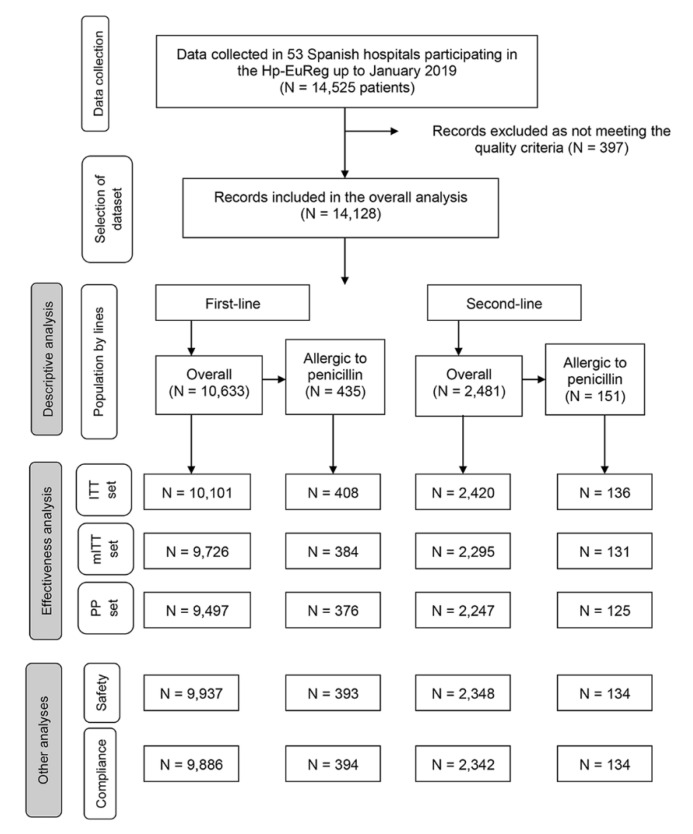
Flowchart of the Spanish patients participating in the Hp-EuReg. N: number of patients included. ITT: intention-to-treat analysis. mITT: modified intention-to-treat analysis. PP: per protocol analysis.

**Table 1 antibiotics-10-00013-t001:** Baseline characteristics of first- and second-line treatments.

Variables		Overall N (%)	1st Line N (%)	2nd Line N (%)
		14,128	10,633	2481
Gender	Female	8795 (62)	6477 (61)	1632 (66)
	Male	5320 (38)	4146 (39)	847 (34)
Age, mean (standard deviation)		50 ± 15	51 ± 14.7	50 ± 14.5
Penicillin allergy	Presence	644 (4.6)	435 (4.1)	151 (6.1)
Indication	Dyspepsia	9152 (65)	6823 (64)	1656 (67)
	Ulcer disease	2103 (15)	1554 (15)	393 (16)
	Others	2868 (20)	2251 (21)	432 (17)
Diagnostic tests	Required endoscopy	8180 (58)	6672 (63)	1061 (43)
Culture	Performed	NA	366 (3.4)	NA
	No resistance		199 (54)	
	Clarithromycin R		52 (14)	
	Metronidazole R		93 (25)	
	Clarithromycin and metronidazole R		18 (4.9)	
	Levofloxacin R		63 (17)	
Treatment length	7 days	182 (1.3)	160 (1.5)	17 (0.7)
	10 days	8615 (61)	6670 (63)	1382 (56)
	14 days	5273 (37)	3764 (35)	1074 (43)
	Others	58 (0.4)	39 (0.4)	8 (0.3)
Proton pump inhibitor dose	Low	5110 (37)	4063 (39)	751 (31)
	Standard	3290 (24)	2605 (25)	458 (19)
	High	5496 (39)	3834 (36)	1239 (50)
Compliance	No (<90% drug intake)	415 (2.9)	304 (3)	69 (2.8)
	Yes (≥90% drug intake)	13,159 (93)	9932 (93)	2302 (92.8)
	Unknown	554 (3.9)	397 (4)	110 (4.4)

N, total of patients included; %, proportion of patients included; R, resistance; NA, not applicable; Low, ≈20 mg omeprazole equivalents b.i.d.; Standard, ≈40 mg omeprazole equivalents b.i.d.; High, ≈60 mg omeprazole equivalents b.i.d.

**Table 2 antibiotics-10-00013-t002:** Effectiveness, safety and compliance of the most frequent therapies prescribed in the first- and second-line treatments.

	Effectiveness						Adverse Events		Compliance	
Treatment	ITT		mITT		PP					
	N (%)	95% CI	N (%)	95% CI	N (%)	95% CI	N (%)	95% CI	N (%)	95% CI
**1st line**	10,101 (83.5)	83–84	9726 (88)	88–89	9497 (89)	88–89	9937 (25)	25–26	9886 (97)	96–97
PPI + C + A + M (Conc)	3996 (86)	85–87	3880 (90)	89–91	3781 (90)	89–91	3963 (28)	26–29	3942 (97)	96–97
PPI + C + A	2712 (78)	76–80	2544 (83)	82–85	2498 (84)	82–85	2617 (15)	13–16	2598 (98)	97–98
PPI + Single–capsule	1574 (88)	86–89	1540 (95)	94–96	1514 (96)	95–97	1566 (25)	23–27	1562 (97)	96–98
PPI + Bi + C + A	1034 (88)	86–90	1015 (91)	89–93	1002 (91)	89–93	1019 (40)	37–44	1021 (98)	98–99
PPI + C + A + M (Seq)	230 (79)	73–84	222 (81.5)	76–86	192 (84)	79–89	230 (49)	42–55	222 (86.5)	81–91
PPI + C + M	124 (59)	50–68	113 (65)	55–73	112 (65)	65–74	119 (16)	10–24	118 (97.5)	93–99
**2nd line**	2420 (79)	77–80	2295 (84)	82–85	2247 (84)	82–86	2348 (28)	27–30	2342 (97)	96–98
PPI + L + A	944 (74)	71–77	893 (78.5)	76–81	881 (79)	76–82	919 (26)	23–29	908 (99)	98–99
PPI + Bi + L + A	463 (86)	83–89	451 (89)	86–92	435 (90)	87–93	454 (33)	28–37	459 (95)	93–97
PPI + Single–capsule	443 (80)	76–84	409 (88)	85–91	398 (89)	85–92	422 (31)	27–36	420 (96)	94–98
PPI + Mx + A	135 (87)	80–92	129 (91)	84–95	129 (91)	84–95	134 (19)	13–27	133 (99)	96–100
PPI + C + A + M (Conc)	120 (74)	65–82	110 (82)	73–89	109 (82)	73–88	112 (24)	17–33	112 (98)	94–100

ITT, intention-to-treat; mITT, modified intention-to-treat; PP, per protocol; N, total of patients included; %, proportion of patients presenting effectiveness/adverse events/compliance; CI, confidence interval; PPI, proton pump inhibitor; C, clarithromycin; A, amoxicillin; M, metronidazole; Single-capsule, three-in-one single capsule; Bi, bismuth; L: levofloxacin; Mx, moxifloxacin; Conc, concomitant administration; Seq, sequential administration.

**Table 3 antibiotics-10-00013-t003:** Multivariate analysis of patients treated with a first-line therapy.

Variables		Overall		PPI + C + A + M (Conc)		PPI + C + A		PPI + Single-Capsule		PPI + Bi + C + A		PPI + C + A + M (Seq)	
		OR (95% CI)	*p*-Values	OR (95% CI)	*p*-Values	OR (95% CI)	*p*-Values	OR (95% CI)	*p*-Values	OR (95% CI)	*p*-Values	OR (95% CI)	*p*-Values
Gender	[R: Female]	1		1		1		1		1		1	
	Male	1.22 (1.07–1.40)	0.004	1.39 (1.11–1.75)	0.004	1.26 (1.00–1.59)	0.050	0.83 (0.51–1.34)	0.442	1.10 (0.79–1.74)	0.690	1.85 (0.84–4.04)	0.125
Age (years)	[R: 18–30]	1		1		1		1		1		1	
	31–50	1.15 (0.92–1.44)	0.232	1.32 (0.92–1.90)	0.136	1.13 (0.77–1.65)	0.530	0.73 (0.24–2.18)	0.568	0.86 (0.38–1.95)	0.723	0.33 (0.07–1.57)	0.163
	51–70	1.08 (0.86–1.35)	0.514	1.23 (0.86–1.77)	0.265	1.10 (0.75–1.61)	0.615	0.72 (0.24–2.12)	0.547	0.82 (0.37–1.85)	0.634	0.32 (0.07–1.62)	0.170
	≥71	1.18 (0.87–1.59)	0.289	1.29 (0.78–2.12)	0.327	1.26 (0.76–2.08)	0.369	0.70 (0.20–2.47)	0.575	0.65 (0.24–1.79)	0.406	0.23 (0.03–1.53)	0.128
Presence of ulcer	[R: No]	1		1		1		1		1		1	
	Yes	1.23 (1.01–1.49)	0.042	1.15 (0.83–1.60)	0.389	1.48 (1.07–2.04)	0.019	1.02 (0.50–2.10)	0.950	3.12 (0.96–10.2)	0.059	0.93 (0.36–2.41)	0.875
Length (days)	[R: 7]	1		NA		1		NA †		NA		NA ‡	
	10	4.46 (3.20–6.23)	0.000	1		2.78 (1.92–4.04)	0.000			1			
	14	4.11 (2.88–5.87)	0.000	1.28 (0.98–1.66)	0.068	2.46 (1.60–3.77)	0.000			0.71 (0.06–8.54)	0.790		
PPI dose of OE	[R: Low]	1		1		1		1		1		1	
	Standard	1.42 (1.21–1.66)	0.000	0.98 (0.75–1.29)	0.888	2.14 (1.67–2.74)	0.000	1.50 (0.84–2.69)	0.168	6.69 (1.35–33.3)	0.020	1.99 (0.24–16.4)	0.521
	High	2.05 (1.72–2.44)	0.000	1.59 (1.24–2.03)	0.000	3.54 (2.49–5.03)	0.000	1.97 (1.08–3.59)	0.027	2.26 (0.63–8.08)	0.211	0.56 (0.22–1.49)	0.244
Compliance	[R: <90% DI]	1		1		1		1		1		1	
	≥90% DI	4.07 (3.04–5.45)	0.000	3.41 (2.14–5.43)	0.000	7.12 (3.69–13.7)	0.000	16 (6.99–36.6)	0.000	1.78 (0.37–8.45)	0.469	2.98 (1.28–6.94)	0.011

Overall, the population receiving a first-line therapy; PPI, proton pump inhibitor; C, clarithromycin; A, amoxicillin; M, metronidazole; Single-capsule, three-in-one single capsule; Bi, bismuth; Conc, concomitant administration; Seq, sequential administration; OR, odds ratio; CI, confidence interval; R, category of reference used for the logistic regression; OE, omeprazole equivalent; DI, drug intake; NA, not applicable; † 99.7% of the patients received 10-days of therapy so comparison in length terms was not possible; ‡ 99.6% of the patients received 10-days of treatment so comparison in terms of length was not possible.

**Table 4 antibiotics-10-00013-t004:** Multivariate analysis of patients treated with a second-line therapy.

Variables		Overall		PPI + L + A		PPI + Bi + L + A		PPI + Single-Capsule	
		OR (95% CI)	*p*-Values	OR (95% CI)	*p*-Values	OR (95% CI)	*p*-Values	OR (95% CI)	*p*-Values
Gender	[R: Female]	1		1		1		1	
	Male	1.39 (1.07–1.81)	0.014	1.03 (0.70–1.50)	0.898	2.83 (1.33–6.05)	0.007	0.96 (0.47–1.93)	0.898
Age	[R: 18–30]	1		1		1		1	
	31–50	0.60 (0.36–0.99)	0.045	0.81 (0.39–1.70)	0.582	0.64 (0.17–2.34)	0.506	0.33 (0.07–1.52)	0.156
	51–70	0.44 (0.27–0.73)	0.001	0.47 (0.23–0.97)	0.041	0.36 (0.10–1.29)	0.118	0.38 (0.08–1.73)	0.211
	≥71	0.37 (0.20–0.69)	0.002	0.43 (0.18–1.03)	0.059	0.24 (0.05–1.09)	0.064	0.73 (0.09–5.82)	0.766
Presence of ulcer	[R: No]	1		1		1		1	
	Yes	1.07 (0.74–1.54)	0.729	1.04 (0.63–1.73)	0.869	1.97 (0.44–8.76)	0.374	0.79 (0.32–1.94)	0.600
Previous C	[R: No]	1		1		1		NA †	
	Yes	0.63 (0.33–1.21)	0.167	0.82 (0.16–4.12)	0.809	0.21 (0.03–1.79)	0.155		
Length	[R: 10]			1		1		1	
	14	1.51 (1.11–2.05)	0.009	3.88 (2.24–6.71)	0.000	3.12 (0.35–27.6)	0.307	0.64 (0.07–5.93)	0.692
PPI dose of OE	[R: Low]	1		1		1		1	
	Standard	1.21 (0.88–1.65)	0.241	1.20 (0.81–1.79)	0.360	2.50 (0.40–15.5)	0.325	1.45 (0.56–3.77)	0.443
	High	1.88 (1.37–2.59)	0.000	1.66 (0.99–2.77)	0.055	3.24 (1.09–9.59)	0.034	1.43 (0.72–2.87)	0.310
Compliance	[R: <90% DI]	1		1		1		1	
	≥90% DI	3.43 (1.71–6.88)	0.001	5.47 (1.27–23.5)	0.023	3.01 (0.91–9.99)	0.071	4.46 (1.02–19.5)	0.047

Overall, the population receiving a second-line therapy; PPI, proton pump inhibitor. L, levofloxacin; A, amoxicillin; Bi, bismuth; Single-capsule, three-in-one single capsule; OR, odds ratio; CI, confidence interval; R, category of reference used for the logistic regression; C, clarithromycin; OE, omeprazole equivalent; DI, drug intake; NA, not applicable; †96% of the patients had previously received clarithromycin so comparison between both groups was not possible.

## Data Availability

MDPI Research Data Policies.

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
