# Peer review of "European Registry on Helicobacter pylori Management: Effectiveness of First and Second-Line Treatment in Spain"

_antibiotics, 2020, doi:10.3390/antibiotics10010013_

Round 1

Reviewer 1 Report

Dear authors and editor,

The manuscript  titled ‘’ European Registry on Helicobacter pylori Management: Effectiveness of first and second-line treatment in Spain. ‘’ o evaluate the effectiveness of first and second-line H. pylori eradication in Spain. It is a multi-center project.

Authors analysed data from 14,128 patients. The standard triple therapy shows an inadequate effectiveness. The best effectiveness in first-line was bismuth single-capsule(period for 10 days), and bismuth-clarithromycin quadruple therapies(period for 14 days). In second-line treatment the best effectiveness was seen in bismuth single-capsule regimen(period for 10 days), and uinolone-containing therapies, with or without bismuth (period for 14 days). The obtained results is in line with the recommendations of Spanish gastroenterologists and in line with current guidelines.

In my opinion these findings is an important significance for individualized treatment as well as to increase effectiveness  of H. pylori eradication in Spain.

From a technical point of view, the manuscript is well-organised, the language is correct and the content is understandable. Statistical tests correctly selected. Literature properly selected and up to date. I believe they add some contribution to the literature.

In conclusion, I support publication of the presented article.

Thank you for your choice me as a reviewer.

Author Response

Thank you very much for your review. 

Reviewer 2 Report

The manuscript is well written. 

Aim of this study was to evaluate the
effectiveness of first and second-line H. pylori treatment in Spain, where
the empirical prescription is recommended.  Overall, 53
Spanish hospitals were included, and 10,267 patients received a first-line
therapy. The best results were obtained with the 10-day bismuth
single-capsule therapy (95% cure rate by intention-to-treat) and with both
the 14-day bismuth-clarithromycin quadruple
(PPI-bismuth-clarithromycin-amoxicillin, 91%) and the 14-day non-bismuth
quadruple concomitant (PPI-clarithromycin-amoxicillin-metronidazole, 92%)
therapies. Second-line therapies were prescribed to 2,448 patients, with
most-effective therapies being the triple quinolone
(PPI-amoxicillin-levofloxacin/moxifloxacin) and the bismuth-levofloxacin
quadruple schemes (PPI-bismuth-levofloxacin-amoxicillin) prescribed for 14
days (92%, 89% and 90% effectiveness), and the bismuth single-capsule (10
days, 88.5%). Compliance, longer duration and higher acid inhibition were
associated with higher effectiveness. “Optimized” H. pylori therapies
achieve over 90% success in Spain.

The methods are adequate.

The results justify the conclusions drawn.

Author Response

Thank you very much for your review. 

Reviewer 3 Report

1) In Introduction: Delete lines from 128 to 135 because it's part of the writing draft

2) Add in line 137, the correlation between H. Pylori and b-defensine, too. So change the period with :

"involved in important diseases such as chronic gastritis, peptic ulcer disease, beta-defensine and skin, or gastric cancer."

3) Add another reference in [1-4]: Pero R. et al.  Beta-defensins and analogs in Helicobacter pylori infections: mRNA expression levels, DNA methylation, and antibacterial activity. Plos One. 2019; 14 (9): e0222295

4) In Table 1 insert in the gender the number of male considered in the study, specify the difference between male and female

5) On line 451 add volume and issue in the reference. They are volume 35 and issue (7)

6) On line 460 change the reference the first name is Gisbert JP and al. "III Spanish Conference on Helicobacter pylori infection". Gastroenterol Hepatol 2013; 36 (6): 375-381

7) On line 487 the reference isn't correct: pages are 140-144

Author Response

Dear reviewer,

We want to thank your revision of the manuscript we have written and the comments you have thought of and stated. We have found them very enriching and we believe they can improve considerably our work.

We are answering each point suggested separately.  In the following sections:

1) In Introduction: Delete lines from 128 to 135 because it's part of the writing draft.

Thank you, we made a mistake adapting our manuscript to the format required.

2) Add in line 137, the correlation between H. Pylori and b-defensine, too. So change the period with: "involved in important diseases such as chronic gastritis, peptic ulcer disease, beta-defensine and skin, or gastric cancer."

We find the relationship of beta-defensines and Helicobacter pylori very interesting, as this provides a more comprehensive idea of the association between these bacteria, the immune response towards it, and chronic gastritis and gastric carcinogenesis.

However, considering that the main aim of this manuscript is to evaluate the different antibiotic-strategies used to treat H. pylori infection specifically in Spain (in order to design the best strategy to be used empirically), we believe that specific issues considering the etiopathogenic role of this bacterium in the human being, exceeds, in our opinion, the content searched. This could lead to the distraction of the readers from the main aim of the study.

Nevertheless, we have added the reference suggested in the next point but without mentioning specifically beta-defensines, as we believe this topic is already included in the list of gastric pathology described. No specific mentioning has been made neither for skin and H. pylori due to the same reason. This fact is included in the “extragastric diseases” title mentioned in lines 130-131.

3) Add another reference in [1-4]: Pero R. et al.  Beta-defensins and analogs in Helicobacter pylori infections: mRNA expression levels, DNA methylation, and antibacterial activity. Plos One. 2019; 14 (9): e0222295.

Thank you for your suggestion. We have added it in the line 131 (Reference 5 in lines 448-449).

4) In Table 1 insert in the gender the number of male considered in the study, specify the difference between male and female.

5,320 males were included in total in the Registry, 4,146 were naive to treatment and 847 received a second line therapy against H. pylori infection.

5) On line 451 add volume and issue in the reference. They are volume 35 and issue (7).

Thank you. It is modified as suggested in line 451.

6) On line 460 change the reference the first name is Gisbert JP and al. "III Spanish Conference on Helicobacter pylori infection". Gastroenterol Hepatol 2013; 36 (6): 375-381.

The article mentioned in Reference 12 (Molina-Infante J, Gisbert JP. Update on the efficacy of triple therapy for Helicobacter pylori infection and clarithromycin resistance rates in Spain (2007-2012). Gastroenterol Hepatol 2013;36(6):375-81, has been suggested to be changed.

However, this article sumarises the most updated evidence concerning clarithromycin resistance rates specifically in Spain, which proove to be higher than the number suggested in year 2008. This increasing rate illustrates the lack of an adequate therapeutic response to the triple standard therapy (only 70% of effectiveness in Spain in that article) and therefore justifies why its use should be abandoned.

The reference suggested by the reviewer also mentions the lack of effectiveness of the triple standard therapy; however, information concerning resistance to clarithromycin is scarce. In addition to this, we have already included the next Spanish Consensus Conference on H. pylori (number IV and last, at the moment) with number reference being 9, which is the updated version of this Consensus meeting, and therefore better adapted to what is established nowadays. In fact, we have added this reference in line 143, following the suggestion of the reviewer.

7) On line 487 the reference isn't correct: pages are 140-144.

We have rechecked this reference and the pages of the document go from I40 to I44, it is not a 1 but an “I”.  

8) Other changes performed:

- I have added the word “performed” in Table 1, to clarify that we are mentioning here the presence of allergy to penicillin.

- I have added the word “Adverse events” before “AE” in line 176, as it had not been previously stated what was referenced by that acronym. On the opposite, this clarification has been eliminated from line 347.

- I have changed the order of the Supplementary Files S2 and S3 in the Supplementary analysis document: file S2 is entitled “serious adverse events in first- and second-line therapies” and file S3 “Proton pump inhibitor (PPI) categories: low, standard and high acid inhibition”. There was an error and it was mentioned before S3 than S2 in the text so it is solved. Lines 178 and 206 now mention File S2 and line 352 File S3.

- I have changed the order of the punctuation sign en relation with the reference included (line 321).

- I have deleted the Spanish title included in References 12 and 9.

- I have completed Reference 13 (line 464), as it was incomplete.

- I have completed Reference 35 (line 518), as it has recently been published the complete information.

Thank you and best regards.